# Graph Convolution Network based Recommender Systems: Learning Guarantee and Item Mixture Powered Strategy

**Leyan Deng**[†], **Defu Lian** [†§*], **Chenwang Wu**[†], **Enhong Chen**[†§]

[†] School of Data Science, University of Science and Technology of China

[§] School of Computer Science and Technology, University of Science and Technology of China

{dleyan, wcw1996}@mail.ustc.edu.cn, {liandefu, cheneh}@ustc.edu.cn

## Abstract

Inspired by their powerful representation ability on graph-structured data, Graph Convolution Networks (GCNs) have been widely applied to recommender systems, and have shown superior performance. Despite their empirical success, there is a lack of theoretical explorations such as generalization properties. In this paper, we take a first step towards establishing a generalization guarantee for GCN-based recommendation models under inductive and transductive learning. We mainly investigate the roles of graph normalization and non-linear activation, providing some theoretical understanding, and construct extensive experiments to further verify these findings empirically. Furthermore, based on the proven generalization bound and the challenge of existing models in discrete data learning, we propose Item Mixture (IMix) to enhance recommendation. It models discrete spaces in a continuous manner by mixing the embeddings of positive-negative item pairs, and its effectiveness can be strictly guaranteed from empirical and theoretical aspects.

## 1   Introduction

With the explosive growth of online information, Internet users rely on recommender systems to alleviate such information overload [1, 2, 3, 4, 5, 6]. In order to take full advantage of the rich graph structure with high-hop neighbors in recommender systems, recent studies have begun to apply Graph Convolution Networks (GCNs) [7] to recommender systems. As expected, GCN-based models have emerged as one of the most attractive approaches to recommender systems due to their powerful graph representation abilities. However, these works only demonstrate the effectiveness of GCNs empirically and do not provide theoretical guarantees.

**GCN-based Recommendation.** [8] was the first to introduce GCN into recommender systems. Specifically, GCN in their approach was applied to the user-user and item-item graphs to learn auxiliary information, respectively. GC-MC [9] proposed to apply GCN to the user-item interaction graph to learn the relations between users and items. SpectralCF [10] directly discovered all possible connectivity between users and items from the spectral domain of the user-item graph. However, SpectralCF fails to capture the information in high order. NGCF [11] further proposed employing the element-wise product between users and embeddings to augment the item features that users care about and the user features that items are selected by. Some works [11, 12, 13, 14, 15] observed that removing the non-linear activation not only simplifies the update operation, but also retains or enhances the recommendation performance. LightGCN [12] further removed the feature transformation to simplify the design of GCN. However, these changes only depend on the empirical

---

[*] Corresponding author.

36th Conference on Neural Information Processing Systems (NeurIPS 2022).

results, without theoretical guarantees. In addition to the user-item interaction matrix, there exist studies combining the social network [16] and knowledge graph [17] to enhance the representations of users and items. In this paper, we mainly consider the user-item interaction matrix, i.e. bipartite graph.

**Generalization Analysis for GNNs.** Due to the numerical success [18, 8, 19], several theoretical studies have been devoted to establishing generalization guarantees for Graph Neural Networks (GNNs). In inductive learning, [20] studied the generalization capability of GNNs by the VC-dimension, which is polynomial with respect to (w.r.t.) the number of parameters and the number of nodes in the graph. [21] provided the first data-dependent bounds via Rademacher Complexity for message passing GNNs and did not allow the analysis of graph information. Subsequently, [22] derived a tighter bound for GNNs based on the PAC-Bayes framework. In transductive learning, [23] analyzed the uniform stability of (single layer) GCNs and thereby deriving their generalization guarantees. They proved that the graph filter, which makes the largest absolute eigenvalue independent of the graph size, satisfies this uniform stability criterion. [24] derived a Rademacher complexity based bound for two layers GCN model. [25] provided a more informative bound according to the relations between normalized graph and input feature at the first layer. However, these works mentioned above only focused on node or graph classification tasks, different from recommendations. The recent works [26, 27] have studied the link prediction and graph embedding tasks; however, they did not concern with the specific models.

**Contributions and paper structure.** The main contributions are shown as follows:

1) We quantify the model complexity for GCN-based recommender systems on the bipartite graph via $l_\infty$-covering number. And then, in Section 3, we relate the Covering number to the generalization error under inductive and transductive settings, leading to the generalization guarantees.

2) Motivated by the proven bound that generalization needs to compromise empirical risk and model complexity and the fact that the recommendation model overfits the sampled discrete data, we propose a simple but effective augmentation strategy, Item mixture (IMix) in Section 4. It makes discrete sample space continuous by mixing positive-negative item pairs about the same user. Notably, we theoretically demonstrate that it is beneficial for recommendation generalization.

3) In Section 5, extensive evaluations are conducted on two datasets, Gowalla and Yelp2018. First, to confirm the previous theoretical foundation, we present the corresponding numerical results, mainly including graph normalization and non-linear activation. Second, we validate the effectiveness of the proposed IMix on different models, and the experimental results consistently show that the recommendation performance is significantly improved after configuring IMix.

We conclude in Section 6. All proofs and more experimental results are provided in the appendix.

## 2 Preliminary

In this section, we first formulate the problem of collaborative filtering (CF) based recommender system. We then describe the GCN-based CF models in detail.

### 2.1 Problem Formulation

We use $u \in \mathcal{U}$ and $i \in \mathcal{I}$ to denote the user and item, and use $\mathcal{R} = \{(u,i)|u \in \mathcal{U}, i \in \mathcal{I}\}$ to denote the collected user-item interactions. Let $\mathcal{Z} = \{(u,i)|u \in \mathcal{U}, i \in \mathcal{I}\}$ to be the set of user-item pairs, for any triplet $(z, z', y)$, $y = +1$ if the preference between user-item pair $z \in \mathcal{Z}$ is higher than $z' \in \mathcal{Z}$, otherwise $y = -1$. Let $f \in \mathcal{F} : \mathcal{U} \times \mathcal{I} \to \mathbb{R}$ be a recommender system. For any user-item pair $z \in \mathcal{Z}$, the model predicts a preference score $f(z)$ by the learned user representation $\boldsymbol{e}_u$ and item representation $\boldsymbol{e}_i$. In this paper, we consider the inner-product as the preference function, i.e., $f(z) = \boldsymbol{e}_u^T \boldsymbol{e}_i$. To optimize the recommendation ability of $f$, we adopt a pairwise ranking loss as follows, for any $(z, z', y)$,

$$loss = l(y(f(z) - f(z'))).$$

## 2.2 GCN-based Recommendation

Let $\boldsymbol{x}_u, \boldsymbol{x}_i \in \mathbb{R}^d$ be the user and item feature, where $d$ is the input dimension. Without loss of generality, we fix the collected user-item interactions as matrix $\boldsymbol{R} \in \mathbb{R}^{|\mathcal{U}| \times |\mathcal{I}|}$, where each entry $\boldsymbol{R}_{ui} = 1$ if user $u$ has interacted with item $i$. Let $\boldsymbol{A}$ and $\tilde{\boldsymbol{A}}$ to be the adjacency matrix and the corresponding laplacian matrix, which are both block anti-diagonal matrices since the graph is bipartite. We formalize these two matrices as follows.

$$\boldsymbol{A} = \begin{bmatrix} \boldsymbol{0} & \boldsymbol{R} \\ \boldsymbol{R}^T & \boldsymbol{0} \end{bmatrix} \text{ and } \tilde{\boldsymbol{A}} = \begin{bmatrix} \boldsymbol{0} & \tilde{\boldsymbol{R}}_u \\ \tilde{\boldsymbol{R}}_i^T & \boldsymbol{0} \end{bmatrix}.$$

Following [28], the existing GCN-based collaborative filtering models can be generalized into two steps. Note that we omit the update formula of items for the sake of brevity.

**(1) Update operation.** The most straightforward layer-wise updations are inspired by the GNN techniques, and they can be summarized as follows,

$$\mathbf{h}_u^{l+1} = \phi \left( \mathbf{W}_1^l \mathbf{h}_u^l + \sum_{i \in \mathcal{N}_u} \tilde{a}_{ui} \mathbf{W}_1^l \mathbf{h}_i^l \right), \tag{1}$$

where $\tilde{a}_{ui} = [\widetilde{\boldsymbol{R}_u}]_{ui}$ represents the assigned weights of user $u$ to item $i$; $\mathcal{N}_u$ is the set of adjacent items of user $u$. Some recent works [11] further adopt the element-wise product between nodes with their neighbors to leverage the consistency between user preferences and item attributes. This operation is formulated as,

$$\mathbf{h}_u^{l+1} = \phi \left( \mathbf{W}_1^l \mathbf{h}_u^l + \sum_{i \in \mathcal{N}_u} \tilde{a}_{ui} \left( \mathbf{W}_1^l \mathbf{h}_i^l + \mathbf{W}_2^l \left( \mathbf{h}_i^l \odot \mathbf{h}_u^l \right) \right) \right). \tag{2}$$

In this paper, we focus on the more complex operation defined in Eq. (2) since the Eq. (1) can be seen as a special case of it when $\boldsymbol{W}_2^l = \boldsymbol{O}$.

**(2) Final node representation.** The CF-based recommender systems require the representation of users and items for the final prediction task. The mainstream approaches include linear combination and concatenation, respectively defined as,

$$\text{Linear combination:} \qquad \boldsymbol{e}_u = \sum_{l=0}^{L} \alpha^l \mathbf{h}_u^l,$$

$$\text{Concatenation:} \qquad \boldsymbol{e}_u = \mathbf{h}_u^0 \oplus \mathbf{h}_u^1 \oplus \cdots \oplus \mathbf{h}_u^L,$$

where $\alpha^l$ is a learnable parameter or hyper-parameter, $L$ is the network depth. In this paper, for convenience, we only consider the embeddings of the last layer as the final representation, i.e., $\boldsymbol{e}_u = \mathbf{h}_u^L$. We will prove that the generalization analysis can be easily extended to other integration operations in appendix.

# 3 Generalization for GCN-based Recommender Systems

We first investigate the model complexity of GCN-based recommender system via covering number with specific radius in Section 3.1. Then we link the $l_\infty$−covering number and generalization errors for inductive and transductive learning in Section 3.2, which finally yield the meaningful bounds.

**Assumptions.** To establish the generalization guarantee, let us immediately make some mild assumptions that are easy to implement. For the loss function, assuming $l : \mathbb{R} \to [-B, B]$ is any bounded $C_l$-$lipschitz$ continuous function. For the non-linear activation function, we assume $\phi$ is $C_\phi$-$lipschitz$ continuous with some $C_\phi > 0$, $\phi(0) = 0$. For the weight matrices, we assume that $\mathbf{W}_1^l, \mathbf{W}_2^l \in \mathbb{R}^{d \times d}$ are shared across layers, and have bounded norms $\|\boldsymbol{W}_1^l\|_2 \le B_1$ and $\|\boldsymbol{W}_2^l\|_2 \le B_2$. For the input feature, we assume $\max_{u \in \mathcal{U}} \|\boldsymbol{x}_u\|_2 \le B_u$ and $\max_{i \in \mathcal{I}} \|\boldsymbol{x}_i\|_2 \le B_i$.

## 3.1 Covering Number for GCN-based Recommender Systems

In classical learning theory, the complexity of function class is closely related to the generalization ability [29], and the typical complexity measurements are VC-Dimension [30] and Rademacher

complexity [31]. In this paper, we adopt the covering number [32] to investigate the model complexity of GCN-based recommender systems. Based on the properties of bipartite graphs, we study the complexity by combining the recommendations of GCN-based models for any user-item pairs.

**Lemma 1** (Covering number bound). *Under Assumptions, we further assume that* $\|\mathbf{h}_u^l\|_\infty, \|\mathbf{h}_i^l\|_\infty \leq b$ *for any* $u \in \mathcal{U}$, $i \in \mathcal{I}$, *and* $l = [L]$, *let* $\gamma = \|\widetilde{\boldsymbol{A}}\|_\infty$. *Given a sample set* $S$ *with size* $n$, *the covering number of* $\mathcal{F}$ *over* $S$ *with specific* $\epsilon$ *is bounded as*

$$
\log \mathcal{N}\left(\mathcal{F}_{|S}, \epsilon, \|\cdot\|_\infty\right) \leq d^2 \log\left(1 + \frac{4(\gamma+1)MB_1\sqrt{d}}{\epsilon}\right)\left(1 + \frac{4\gamma b M B_2\sqrt{d}}{\epsilon}\right).
$$

*Moreover, when* $\epsilon \leq 4M\sqrt{d}\max\{(\gamma+1)B_1, \gamma b B_2\}$,

$$
\log \mathcal{N}\left(\mathcal{F}_{|S}, \epsilon, \|\cdot\|_\infty\right) \leq 2d^2 \log \frac{8M(\gamma+1)\sqrt{d B_1 B_2 b}}{\epsilon}, \ where
$$

$$
M = C_\phi \mathcal{C}^{2L-1}(B_u + B_i)^2 \frac{(2\mathcal{C})^L - 1}{2\mathcal{C} - 1},
$$

$$
\mathcal{C} = C_\phi[B_1 + \gamma(B_1 + B_2 b)].
$$

**Remark 1.** The key idea of the proof is to exploit the properties of the bipartite graph in the recommender system. To investigate the complexity of the recommendation for any user-item pair generated by GCN-based models, we establish the connection between the complexities of function space and weight space. Here $\mathcal{C}$ describes the layer-wise updation complexity, which is the important term for the following generalization analysis. According to the norm bound of hidden states, the assumption can be naturally satisfied when the activation function is bounded. And we will discuss the role of unbounded activation in Section 3. Note that the layer-wise complexity reduces to $\mathcal{C} = C_\phi(\gamma+1)B_1$ when ignoring element-wise between users and items, which means that there is no need to discuss the value of $b$.

## 3.2 Generalization Bound for Inductive & Transductive Learning

Transductive and inductive learning are both commonly used in GNNs and recommender systems. The primary difference between these two settings is whether the testing set depends on the training set. The testing samples in inductive learning are drawn from some unknown distribution, while in transductive learning, they are sampled without replacement from a fixed data set. In the following, we first relate the generalization error for two settings to the covering number of GCN-based recommendations, then derive the final generalization guarantees using Lemma 1.

**Inductive Learning.** Let $\mathcal{D}$ to be a fixed but unknown distribution over $\mathcal{Z} \times \mathcal{Z} \times \{-1, +1\}$, we assume that all samples in training set $S_m = \{(z_i, z_i', y_i)\}_{i=1}^m$ are i.i.d. according to $\mathcal{D}$, denoted as $S_m \sim \mathcal{D}^m$, and we aim to make accurate prediction for any unobserved samples $(z, z', y) \sim \mathcal{D}$. Under this inductive setting, the empirical error over the training set $S_m$ and the corresponding generalization error is defined as follows,

$$
\begin{aligned}
\text{Empirical error:} \quad & \hat{\mathcal{L}}_m(f) = \frac{1}{m}\sum\nolimits_{(z_i, z_i', y_i) \in S_m} l(y_i(f(z_i) - f(z_i'))), \\
\text{Generalization error:} \quad & \mathcal{L}(f) = \mathbb{E}_{(z, z', y)\sim\mathcal{D}}[l(y(f(z) - f(z')))].
\end{aligned}
\tag{3}
$$

Under inductive learning, we can derive the following generalization error bound by the complexity term according to the covering number.

**Lemma 2.** *Let* $\mathcal{F}$ *be a real-valued function class taking values in* $[-e, e]$, *and assume that* $\mathbf{0} \in \mathcal{F}$. *Under assumptions, for any function* $f$ *in a class* $\mathcal{F}$, *with probability of at least* $1 - \delta$ *over an i.i.d. size-*$m$ *training set, we have*

$$
\mathcal{L}(f) \leq \hat{\mathcal{L}}_m(f) + 4C_l \inf_{\alpha>0}\left(\frac{4\alpha}{\sqrt{m}} + \frac{12}{m}\int_\alpha^{2e\sqrt{m}}\sqrt{\log\mathcal{N}\left(\mathcal{F}_{|S_m}, \epsilon, \|\cdot\|_2\right)}d\epsilon\right) + 4B\sqrt{\frac{2\log 4/\delta}{m}}.
$$

**Transductive Learning.** For the transductive learning setting, the full samples are observed prior to learning. Given a fixed set $S = \{(z_i, z_i', y_i)\}_{i=1}^{m+u}$, a labeled training set $S_m$ with size $m$ is selected from $S$ uniformly at random. The goal of transductive learning is to predict the labels of the samples

in testing set $S_u = S \backslash S_m$. Under this transductive setting, the formulation of empirical error $\hat{\mathcal{L}}_m(f)$ is similar to Eq. (3), and the generalization error is defined as follows,

$$\mathcal{L}_u(f) = \frac{1}{u} \sum_{(z_i, z_i', y_i) \in S_u} l(y_i(f(z_i) - f(z_i'))).$$

Following the same strategy adopted in proving Lemma 2, we again need to perform a generalization bound under transductive setting via covering number of function class. As shown above, the generalization and empirical error under transductive setting do not depend on any underlying distributions. The important difference induces two versions of generalization analysis. For the transductive learning, we consider the transductive Rademacher complexity [33] defined on both the training and testing data. Then the covering number bounded generalization error can be derived by the interdependence between model complexity measurements [34, 35].

**Lemma 3.** *Let $\mathcal{F}$ be a real-valued function class taking values in $[-e, e]$, and assume that $\mathbf{0} \in \mathcal{F}$. Let $Q_1 = \frac{1}{u} + \frac{1}{m}, Q_2 = \frac{m+u}{(m+u-1/2)(1-1/2(\max(m,u)))}$ and $c_0 < 5.05$. Under assumptions, for any function $f$ in a class $\mathcal{F}$, with probability of at least $1 - \delta$ over random partitions of $S$, we have*

$$\mathcal{L}_u(f) \leq \hat{\mathcal{L}}_m(f) + 2C_l \inf_{\alpha > 0} \left( \frac{4\alpha}{\sqrt{m+u}} + \frac{12}{m+u} \int_\alpha^{2e\sqrt{m+u}} \sqrt{\log \mathcal{N}\left(\mathcal{F}_{|S}, \epsilon, \|\cdot\|_2\right)} d\epsilon \right)$$
$$+ Bc_0 Q_1 \sqrt{\min(m,u)} + 2B\sqrt{\frac{Q_1 Q_2}{2} \ln \frac{1}{\delta}}.$$

Armed with Lemma 2, Lemma 3 and Lemma 1, we now present the generalization error bounds for inductive and transductive learning.

**Proposition 1** (Generalization Bound). Under assumptions, for any function $f$ in a class $\mathcal{F}$, in inductive learning, with probability of at least $1 - \delta$, we have,

$$\mathcal{L}(f) \leq \hat{\mathcal{L}}_m(f) + \frac{24C_l}{\sqrt{m}} \mathcal{C}^{2L}(B_u + B_i)^2 d\sqrt{2\log\left(8mM(\gamma+1)\sqrt{dB_1 B_2 b}\right)} \tag{4}$$
$$+ \frac{16C_l}{m} + 4B\sqrt{\frac{2\log 4/\delta}{m}}.$$

Accordingly, in transductive learning, with probability of at least $1 - \delta$, we have,

$$\mathcal{L}_u(f) \leq \hat{\mathcal{L}}_m(f) + \frac{24C_l}{\sqrt{m+u}} \mathcal{C}^{2L}(B_u + B_i)^2 d\sqrt{2\log\left(8(m+u)M(\gamma+1)\sqrt{dB_1 B_2 b}\right)}$$
$$+ \frac{4C_l \sqrt{2mu}}{(m+u)^2} + Bc_0 Q_1 \sqrt{\min(m,u)} + 2B\sqrt{\frac{Q_1 Q_2}{2} \ln \frac{1}{\delta}}.$$

**Remark 2.** Proposition 1 states that the gap between generalization error and empirical error converges uniformly to 0 with the rate of $\sqrt{(\log m)/m}$. We first discuss several obvious findings about the dependency of the bound on some terms. For the hidden dimension $d$, the generalization bound scales as $\mathcal{O}(d\sqrt{\log d})$; for the input feature norms $B_u$ and $B_i$, the bound scales as $\mathcal{O}\left((B_u + B_i)^2 \sqrt{\log(B_u + B_i)}\right)$. The most important term is the layer-wise complexity $\mathcal{C}$, which depends on the value of $\gamma$ according to the graph structure, the spectral norms of weight matrices $B_1$ and $B_2$, and the value of $C_\phi$ and $b$ according to the non-linear activation function. A natural idea is to constrain the weight matrices during training such that the generalization bound would not grow exponentially with the network depth $L$, which is a regularization techniques commonly used in machine learning. The subsequent discussions focus on the $\gamma$, $C_\phi$ and $b$ to investigate the influence of graph normalization and non-linear activation functions.

**Role of graph normalization.** It is easy to see that $\gamma$ is equal to the maximum degree $D_{\max}$ of the user-item interaction graph without graph normalization. Therefore, fixed other terms in $\mathcal{C}$, the generalization bound grows as $O(D_{\max}^L)$. For the random-walk graph, i.e., $\widetilde{A} = D^{-1}A$, $\gamma = 1$, and for symmetric normalized graph, i.e., $\widetilde{A} = D^{-1/2}AD^{-1/2}$, $\gamma \leq \sqrt{D_{\max}/D_{\min}}$, and hence, the growth is much smaller than the unnormalized graph.

**Role of non-linear activation.** As pointed above, the activation functions affect the generalization performance through two quantities: the Lipschitz constant $C_\phi$ and the norm bound of hidden states $b$. Due to the most activation functions satisfy the condition $C_\phi \leq 1$, and particularly LeakyReLU, ReLU, and Tanh are $1\text{-}Lipschitz$, $C_\phi$ presents negligible influence. For the norm bound $b$, while the bounded activation naturally provides explicit value, the unbounded function can also satisfy the assumption by constraints on the weight matrices. As discussed in Remark 2, weight regularization is widely used in machine learning and can help improve model generalization performance to prevent overfitting. In addition, one recent work [36] gives a new theoretical finding that non-linearity does not bring considerable expressive power to improve representation learning in GNNs, but negatively increase the difficulty of model training. Therefore, this generalization bound provides theoretical support for removing the activation function to simplify the model without incurring a performance penalty.

## 4  Item Mixture Powered Recommendation

The upper bound in Proposition 1 inspires us that the generalization needs to balance empirical risk and model complexity. Most models [37, 11] inevitably lead to excessive complexity in pursuit of high expressiveness, but simple models may lead to higher empirical errors, so it is valuable to reduce the complexity while ensuring model expressiveness. In addition, the recommendation algorithm is trained based on the limited discrete data obtained by sampling, making the model perform well in the discrete space of sampling and prone to overfitting. Considering the above and inspired by Mixup [38], we propose Item Mixture (IMix) to power recommendation, which forces the model to deal with regions between discrete samples continuously by mixing any two positive-negative item pairs. In addition, IMix does not make changes to the model, while we can theoretically prove that it reduces the model complexity, and then enjoys generalization. Below we give specific details.

Since the original representations of items are discrete and unprocessable (e.g., orange or apple), we suggest mixing their embeddings. Specifically, given a triplet $(u, i, i')$ with a label $y_i$, where $y_i = 1$ if user $u$ prefers item $i$ to $i'$, otherwise $y_i = 0$. The embeddings of $(u, i, i')$ are $(e_u, e_i, e_{i'})$ computed according to Section 2.2, we arbitrarily sample another triplet $(u, j, j')$ with label $y_j$ of the same user. If $y_i = y_j$, we can obtain mixed embedding triplet $(e_u, \widetilde{e}_i, \widetilde{e}_{i'}) = (e_u, \lambda e_i + (1 - \lambda)e_j, \lambda e_{i'} + (1 - \lambda)e_{j'})$ with $\lambda \sim D_\lambda = Beta(\alpha, \beta)$, otherwise, $(e_u, \widetilde{e}_i, \widetilde{e}_{i'}) = (e_u, \lambda e_i + (1 - \lambda)e_{j'}, \lambda e_{i'} + (1 - \lambda)e_j)$. So in IMix, we will train the recommendation model on the new mixed embeddings. We consider the logistic loss $\ell(f(x), y) = \log(1 + e^{f(x)}) - yf(x)$, the standard loss and IMix loss are defined as

$$\text{Standard loss:} \qquad \mathcal{L}_m^{std} = \frac{1}{m}\sum\nolimits_{k=1}^{m} \ell(e_{u_k}^T(e_{i_k} - e_{i'_k}), y_k),$$

$$\text{IMix loss:} \qquad \mathcal{L}_m^{IMix} = \frac{1}{m}\sum\nolimits_{k=1}^{m} \ell(e_{u_k}^T(\widetilde{e}_{i_k} - \widetilde{e}_{i'_k}), y_k). \tag{5}$$

Considering the motivation above, IMix preserves the model expressiveness without modification of the model structure. Besides, it transforms the learning of discrete sample pairs into the learning of continuous regions between sample pairs, which may help to alleviate the overfitting.

Notably, although IMix is inspired by the generalization bound in Proposition 1, it is decoupled from the model structure. Therefore, it is theoretically applicable to any embedding-based recommendation model, and we will show its performance in non-GCN-based models in Appendix A.3.

Next, we theoretically guarantee the effectiveness of the proposed IMix. Following a similar approach in [39], we adopt second-order Taylor expansion to state the IMix training loss is approximately equivalent to standard empirical error with a regularization term.

Consider a training set $S_m$, denote by $D_u$ all users and $D_i$ all item-pairs in $S_m$, where $(i, i') \in D_i$ if $(i', i) \in D_i$. We assume that the samplings of user and item-pair are independent, and for $\forall u \in D_u, \forall (i, i') \in D_i, (u, i, i') \in S_m$. For convenience, let $y_{u,i,i'}$ denote the label of $(u, i, i')$.

**Lemma 4.** *Consider the symmetric dataset $S_m$ and denote $\hat{\Sigma} = \frac{1}{m}\sum\limits_{k=1}^{m}(e_{i_k} - e_{i'_k})(e_{i_k} - e_{i'_k})^T$, the second-order approximation of IMix loss defined in Eq. (5) is given by*

$$\mathcal{L}_m^{mix} \approx \mathcal{L}_m^{std} + \mathbb{E}_\lambda(1 - \lambda)^2 \cdot \frac{1}{2m}\sum\nolimits_{k=1}^{m}\left[\frac{e^{\eta_k}}{(1 + e^{\eta_k})^2}e_{u_k}^T\hat{\Sigma}e_{u_k}\right],$$

*where $\eta_k = e_{u_k}^T(e_{i_k} - e_{i'_k})$.*

Then we are ready to investigate the following function class: let $\Sigma = \mathbb{E}_{(i_k, i'_k)}\left[(e_{i_k} - e_{i'_k})(e_{i_k} - e_{i'_k})^T\right]$,

$$\mathcal{F}_\tau^{mix} = \{\mathcal{F}, \text{ such that } \mathbb{E}_{(u_k, i_k, i'_k)}\left[\frac{e^{\eta_k}}{(1 + e^{\eta_k})^2}e_{u_k}^T \Sigma e_{u_k}\right] \leq \tau\}.$$

**Remark 3.** Let $\psi(u) = \frac{e^u}{(1+e^u)^2}$, similar to [39, 40], we assume that the distribution of $(e_i - e_{i'})$ is $\rho-$retentive for some $\rho \in (0, 1/2]$, that is, if for any non-zero vector $e_u$, $\left[\mathbb{E}_{(i,i')}\psi\left(e_u^T(e_i - e_{i'})\right)\right]^2 \geq \rho\min\{1, \mathbb{E}_{(i,i')}\left(e_u^T(e_i - e_{i'})\right)^2\}$. Suppose there exists a loss function, which is $C_l-lipschitz$ continuous and bounded by $[-B, B]$, for any $f \in \mathcal{F}_\tau^{mix}$, with probability of at least $1 - \delta$, we have the following bound on generalization error.

$$\mathcal{L}(f) \leq \hat{\mathcal{L}}(f) + 2C_l\left(\max\{(\frac{\tau}{\rho})^{1/4}, (\frac{\tau}{\rho})^{1/2}\}\sqrt{\frac{rank(\Sigma)}{|D_i|}}\right) + B\sqrt{\frac{2\log 2/\delta}{m}}. \tag{6}$$

If we consider the function class with a general regularization technique on the weight matrices, as we discussed above, the constraint on weights will ultimately control the node representation. Therefore, for the general condition, we focus on the function class $\mathcal{F}_\tau^{std} = \{\mathcal{F}|\mathbb{E}_{(u,i)}\left[\|e_u\|_2^2 + \|e_i\|_2^2\right] \leq \tau\}$, then the similar proof strategy would yield the generalization error bound $\mathcal{L}(f) \leq \hat{\mathcal{L}}(f) + 2C_l\sqrt{\frac{\tau^2}{|D_i|}} + B\sqrt{\frac{2\log 2/\delta}{m}}$, we can conclude that when $rank(\Sigma_X) < \tau$, IMix has a better generalization. It is easily met because $\tau$ tends to be large to pursue high performance. For example, in the experiment, the embedding size $d$ is 64 (noted that $rank(\Sigma_X) < d$), while the minimum value of $\tau$ is 130.8.

## 5 Experiments

In this section, we conduct extensive experiments to validate our theoretical findings and the effectiveness of the proposed IMix.

**Dataset.** We follow existing work [11, 12], using two datasets: (1) Gowalla, contains 1,027,370 check-in information from 29,858 users to 40,981 locations; (2) Yelp2018, comprises 31,668 users who have reviewed 38,048 items (e.g., restaurants, bars) with 1,561,406 times. The training set, validation set, and test set are randomly sampled in a ratio of $7 : 1 : 2$.

**Model Structure.** Two competitive GCN-based algorithms, NGCF [11] and LightGCN [12], are used to construct experiments. Among them, NGCF explicitly models the high-order connectivity between users and items through message construction and message aggregation, thereby improving the expressiveness of embedding. LightGCN, on the other hand, is keenly aware that the nonlinear activation function and feature transformation of NGCF have no specific semantics in the recommendation, so it is simplified to only a domain message aggregation module. Correspondingly, these models are configured with the proposed IMix denoted as NGCF-IMix and LightGCN-IMix, respectively.

We use the public implementation version of LightGCN and follow the default parameters of their algorithms unless otherwise specified. More details are provided in Appendix A.1.

**Metrics.** The recommendation performance is measured by the positive items' rankings in the test set. We use two ranking metrics widely used in recommender systems [41, 42, 43]: Recall and NDCG. Recall at a cutoff K, denoted as Recall@K, denotes the proportion of positive items predicted to be correct. NDCG at a cutoff K, denoted as NDCG@K, measures the reward for the positive items' position in the top-K recommendation list. For these two metrics, we truncate the ranked list K to 20.

### 5.1 Numerical Discussion

In this subsection, we discuss the implication of the generalization bound defined in Proposition. 1, and the results are built on NGCF because it is more similar to traditional GCNs.

---
https://github.com/kuandeng/LightGCN

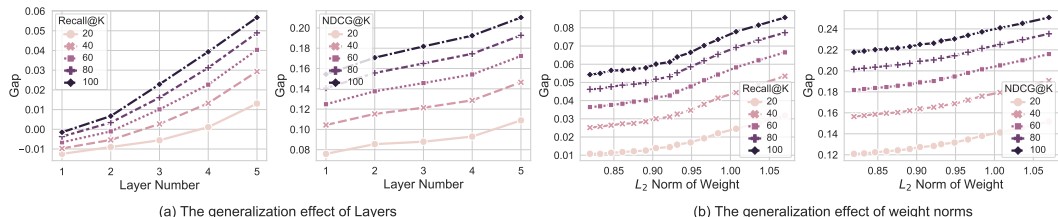

(a) The generalization effect of Layers      (b) The generalization effect of weight norms

Figure 1: (a): the effect of number of layers $L$ on Generalization Performance on Gowalla. (b): the effect of weight norms $B_1$, $B_2$ on Generalization Performance on Gowalla.

**Quantitative Analysis.** We investigate the effects of two major values on the generalization gap, which represents the differences between recommendation performances in training and testing sets. We consider the number of layers $L$, as shown in the left part of Fig. 1. As the number of layers increases, the generalization gaps of ndcg@K and recall@K significantly increase. The empirical results imply that the generalization error is proportional to the number of layers, consistent with the theoretical analysis. We also consider the spectral norms of weight matrices $B_1$ and $B_2$, as shown in the right part of Fig. 1. We compare the generalization abilities with different weight norms. As the norms increase, the generalization gaps increase steadily. Compared with the number of layers, we can observe that the weight norms increase more slowly. The theoretical reason is that the generalization performance is $L$-th order with respect to the weight norms $B_1$ and $B_2$, and exponential with respect to the number of layers $L$.

**The Role of Normalized Graph.** We compare the generalization performances with different normalized graphs, as shown in the left part of Fig. 2. It is clear that the two normalized graphs have higher generalizations than the unnormalized graph. This result is consistent with the theoretical findings. The comparison between two normalized graphs depends on the specific user-item interaction graph.

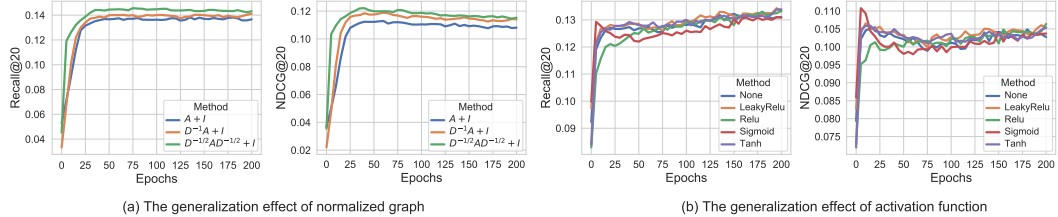

(a) The generalization effect of normalized graph      (b) The generalization effect of activation function

Figure 2: (a): the effect of normalized graph on Generalization Performance on Gowalla. (b): the effect of activation function on Generalization Performance on Gowalla.

**The Role of Non-linear Activation Function.** We study the role of non-linear activation function in GCN-based recommender systems. Since most activation functions are all 1-Lipschitz continuous, we compare the generalization performances with the popular activation functions, including LeakyReLU, ReLU, Sigmoid, and Tanh. Furthermore, we compare these results with the model without activation function. As shown in the right part of Fig. 2, these models present similar performances, which provide theoretical support for removing activation functions. Although the Sigmoid presents great recommendations at first, it overfitted as the training continued, while the performance of other models is still improved, and their optimal performance tends to be unified.

## 5.2 Performance Analysis of IMix

**Effectiveness Validation.** We apply the proposed IMix to NGCF and LightGCN and perform a detailed comparison, as shown in Table 1. The percentage of relative improvement is shown in parentheses. Obviously, the proposed augmentation strategy (corresponding to NGCF-IMix and LightGCN-IMix) significantly improves the performance of the corresponding models (NGCF and LightGCN). For example, in the Gowalla dataset, the best performance of NGCF on Recall@20 is 0.1308, while with IMix configured, the performance improves to an encouraging 0.1496. In addition, the improvements of IMix on NGCF are larger than that on LightGCN. We reasonably believe that

Table 1: Overall evaluation. All models' performance is improved after configured with IMix.

| Dataset | | Gowalla | | Yelp2018 | |
|---|---|---|---|---|---|
| Layer # | Method | Recall@20 | NDCG@20 | Recall@20 | NDCG@20 |
| 1 layer | NGCF | 0.1308 | 0.1032 | 0.0419 | 0.0336 |
| | NGCF-IMix | 0.1451(+10.93%) | 0.1146(+11.05%) | 0.0492(+17.42%) | 0.0360(+7.14%) |
| 2 layer | NGCF | 0.1274 | 0.1050 | 0.0431 | 0.0348 |
| | NGCF-IMix | 0.1454(+14.13%) | 0.1164(+10.86%) | 0.0491(+13.92%) | 0.0391(+12.36%) |
| 3 layer | NGCF | 0.1303 | 0.1082 | 0.0435 | 0.0353 |
| | NGCF-IMix | 0.1496(+14.81%) | 0.1191(+10.07%) | 0.0504(+15.86%) | 0.0405(+14.73%) |
| 1 layer | LightGCN | 0.1556 | 0.1340 | 0.0526 | 0.0427 |
| | LightGCN-IMix | 0.1699(+9.19%) | 0.1436(+7.16%) | 0.0562(+6.84%) | 0.0458(+7.26%) |
| 2 layer | LightGCN | 0.1672 | 0.1425 | 0.0564 | 0.0462 |
| | LightGCN-IMix | 0.1765(+5.56%) | 0.1507(+5.75%) | 0.0621(+10.10%) | 0.0507(+9.74%) |
| 3 layer | LightGCN | 0.1759 | 0.1500 | 0.0602 | 0.0492 |
| | LightGCN-IMix | 0.1793(+1.93%) | 0.1524(+1.6%) | 0.0644(+6.98%) | 0.0527(+7.11%) |

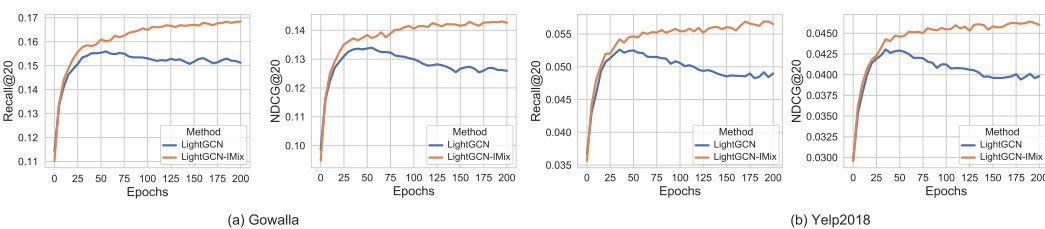

(a) Gowalla       (b) Yelp2018

Figure 3: Test curves (Recall@20 and NDCG@20) for 200 epochs of training.

LightGCN is more expressive (better performance), and the benefit of the boosting strategy decreases accordingly. Nonetheless, the performance on LightGCN is also a pleasant surprise.

Besides, Fig. 3 presents the performance curves regarding Recall and NDCG for the test set during the training of LightGCN-based models (LightGCN and LightGCN-IMix). Throughout the training process, LightGCN-IMix consistently outperforms LightGCN. After 50 epochs, LightGCN shows different degrees of overfitting. On the contrary, the performance of LightGCN-IMix is still further improved, and their gap is gradually enlarged. We reasonably believe that the proposed strategy adds a data-dependent regularization term to LightGCN (Lemma 4), which helps to alleviate overfitting.

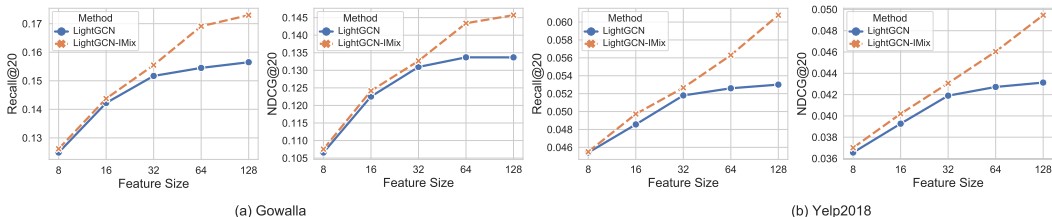

(a) Gowalla       (b) Yelp2018

Figure 4: Performance comparison of different embedding sizes.

**Sensitivity w.r.t. Model Hyperparameters.** We also explore the impact of the model's hyperparameters: different feature sizes (dimension of $x_u$ and $x_i$) and hidden layer sizes (dimension of $W_1^l$ and $W_2^l$). Here, the number of layers is set to 1, and the results are shown in Fig. 4 and Fig. 5.

In Fig. 4, we find that the larger the feature size, the more beneficial it is to the recommendation. Moreover, LightGCN-IMix is better than LightGCN, and even more, the better the performance of the original model, the better the improvement after configuring IMix, which highlights the superiority of the proposed IMix.

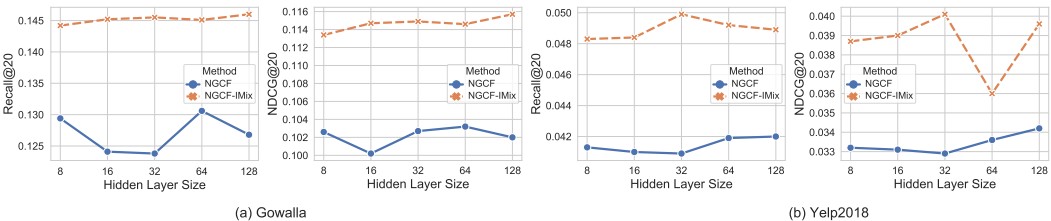

Figure 5: Performance comparison of different hidden layer size.

Fig. 5 reveals that the hidden layer size does not cause a significant change in model performance, while when the hidden layer size is only 8, the performance is sufficiently satisfactory, which provides support for the practicality of GCN-based algorithms. In addition, with similar conclusions to Fig. 4 and Table 1, the IMix-configured models outperform the original models in all cases. In conclusion, we validate the effectiveness of IMix.

## 6  Conclusion and Future Work

GCNs achieve superior empirical success in recommender systems but lack theoretical understanding. In this paper, we provide a model complexity analysis for GCN-based recommendations via Covering number, and then induce the generalization error bounds in inductive and transductive learning. We report several theoretical findings and again verify them via experimental results. Inspired by the generalization analysis, we introduce an augmentation strategy, Item Mixture, to improve the recommendation generalization by mixing item embeddings.

There are several interesting future directions. First, it would be interesting to further analyze the user-item bipartite graph with social relationship and knowledge graph. The second is to extend our analysis to investigate the plausibility of the other GNN-based variants. The third is to study the role of graph spectral properties in generalization ability.

## Acknowledgments and Disclosure of Funding

The work was supported by grants from the National Key R&D Program of China (No. 2021ZD0111801) and the National Natural Science Foundation of China (No. 62022077).

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
