# OpenReview forum: "Graph Convolution Network based Recommender Systems: Learning Guarantee and Item Mixture Powered Strategy"
_NeurIPS.cc/2022/Conference — NeurIPS 2022 Accept_

### Official Review · Reviewer_sKUH · 2022-07-08

**Rating:** 6
**Confidence:** 2
**Soundness:** 3 good
**Presentation:** 2 fair
**Contribution:** 2 fair

**Summary:**

This paper gave out the proof that why graph convolution network is able to improve the performance of recommender system and based on the theoretically analysis it proposes a new method iMix to power recommendation. The proofs focused on the generalization bound adopting the Covering number theory. It firstly exploited the properties of the user-item interaction graph in the recommender system. Then, it states that generalization error and empirical error can be coverage to 0 while supervised data is large enough. In terms of the generalization bound achieved, graph normalization showed its superiority while non-linear activation function was not necessary. iMix is model-agnostic, preserving the model expressiveness by mixing any two positive-negative item pair instead of discrete individual items. And iMix can well satisfy the generalization with a easily-met parameter. Experiment on generalization quantity, role of normalized graph, role of non-linear activation function, performance of iMix and its hyperparameters were conducted, and showed the consistency on proof and analysis.

**Questions:**

1. There should be more optical images or graph that help reader better understand straightforward, e.g., the thought flow, proof flow, instead of full of mathematical equations.
2. For line 120, what does the Z means by? What’s its usage? It seems that it is different from the Z defined in line 72, which is confusing.
3. For line 206, it says it arbitrarily sample another embedding triplet (e_u,e_i,e_i'), whether it should be correctly written as (e_u,e_j,e_j')?
4. For line 230, there is an inequation that can conclude rank(Σ_X )<τ, but referenced to the appendix the inequation is different from the one here. How could that be?
5. For line 231, the relation between τ and γ is confusing. It jumps without explanation.
6. For line 232, it says, in the experiment minimum value of γ is 130.8. According to line 176, γ is equal to the maximum degree of the user-item interaction graph without graph normalization. So how can it be 130.8 in the experiment, especially a decimal rather than an integer?



**Limitations:**

This work focuses on the simple GCN based recommender system, while nowadays there are many other kinds of recommender system involving social relations on users and knowledge graph on items. The potential of suck kinds of recommender system is still not discovered yet, which means whether it is suitable for and outperforms the complex models. The role of graph spectral properties in generalization ability is still known, neither.

**Strengths And Weaknesses:**

Strengths:
a.	The paper gives a bunch of firm evidence for the proof, showing the author’s strong background and well understanding of the previous works.
b.	The flow of the whole paper is intelligible, along with the process of logic and thought.
c.	The work is of the significance since so far there is not any other work gave such complete and theoretical analysis.

Weaknesses:
a.	This paper mostly focuses on the proof of generalization bound and the iMix effectiveness, on the contrary the novelty of new method for recommender is less.
b.	Notations given is in a variety and ambiguous, i.e., d for dimension size and node degree. Their definition and explanation also scatter. Notations table and description of the relations among notations is needed.

---

> ### Author Response · Authors · 2022-08-02
> **Response to Reviewer sKUH**
>
> We would like to sincerely thank Reviewer sKUH for positive evaluations and valuable comments for improvement.
>
> ### Q1. The novelty of new method for recommender is less.
>
> Our contributions might not have been well received, and we restate and emphasize them:
>
> - The first generalization guarantee for GCN-based recommender systems.
> - The effective augmentation strategy IMix and its theoretical guarantee.
> - Promising numerical results on extensive experiments.
>
> For the new method IMix,  first, inspired by the generalization bound of GCN-based recommendation model, it desires to reduce the model complexity while ensuring model expressiveness (proved in **Lemma 4**). Second, one of the main challenges of recommendation is the discreteness and extreme sparsity of data, which makes the model often highly dependent on (overfit or underfit) dataset [R1]. The proposed IMix linearly processes the items’ embeddings, which may improve the embedding space’s vicinal smoothness and reduce the model’s dependence on data (proved in **Fig.3** and **Fig.8**). Notably, we have also shown its potential in non-GCN-based recommendation (**Appendix A.3**).
>
> ### Q2. There should be a notations table and more optical images or graph.
>
> Following your suggestion, we have summarized all notations of the paper, please refer to **Table 2** in the revised appendix. In order to highlight the logical structure of our work, we rephrased Section 3 and briefly summarized proof process before each formal proof in **Appendix B-F**.
>
> ### Q3. Confusing notations.
>
> > **(1) Notations given is in a variety and ambiguous, i.e., d for dimension size and node degree.**
> >
>
> We denote the dimension size by $d$, the maximum node degree by $d_{\max}$ and the minimum node degree by $d_{\min}$. It seems that the repeated use of "d" caused confusion. In the revised paper, we have used a new symbol $D$ to represent node degree.
>
> > **(2) The "Z" in line 72 and the "Z" in line 120 are confusing.**
> >
>
> The two Z have different fonts. The $\mathcal{Z}$ in line 72 is a calligraphic font, representing the set of user-item pairs. And the $Z$ in line 120 is normal font, defined before line 121 to simplify the upper bound in Lemma 1. To avoid confusion, we have replaced the $Z$ with a new symbol $M$ in the revised paper.
>
> > **(3) The embedding triplet $(e_u, e_i, e_{i'})$ should be $(e_u, e_j, e_{j'}).$ The inequation in line 230 is different from the appendix.**
> >
>
> Thank you for your pointing out typos, we have corrected them in the revised paper. For the second typo, the upper bound presented in the main text is correct. As we discussed, IMix achieves better generalization when $rank(\Sigma_{X}) < \tau$, which is easily met in the experiment.
>
> > **(4) The $\tau$ and $\gamma$ is confusing. The minumum value of $\gamma$ is unreasonable.**
> >
>
> We are sorry for these typos. The $\gamma$ in line 231 and line 232 should be $\tau$. Therefore, the minimum value of $\tau$ in the experiment is 130.8 following the definition in line 228. We have fixed these typos in the revised paper.
>
> Thank you again for your careful review, and we are sorry for the confusion caused by our writing mistakes. In the revised paper, we have carefully checked the symbols and corrected the typos.
>
> ### Q4. The potential of such kinds of recommender system is still not discovered yet, whether it is suitable for and outperforms the complex models.
>
> As the reviewer states, there are many other kinds of recommender systems. Notably, with the advantages of handling the structural data and exploring structural information, GCN-based methods have been widely used in various recommendations such as social recommendation [R2,R3], sequence recommendation [R4], and session recommendation[R5], which convinced us of their potential.
>
> Besides, our theoretical analysis provides covering number-based generalization bounds for inductive and transductive learning, which may provide insights for theoretical studies of other recommendation models. Theoretically, IMix could work as long as the model is inference based on embeddings and trained with bounded ranking loss. These scalable potentials make us look forward to generalizing them to develop more connections between various relation-based recommender systems.
>
> ### Q5. The role of graph spectral properties in generalization ability is still unknown.
>
> As we say $l_{\infty}$-Covering number is a worst-case analysis, future work towards $l_1$ or $l_2$-Covering numbers may shed light on the role of graph spectral properties.
>
> [R1] Neural collaborative filtering, WWW2017.
>
> [R2] Group-buying recommendation for social e-commerce, ICDE2021.
>
> [R3] Self-supervised multi-channel hypergraph convolutional network for social recommendation, WWW2021.
>
> [R4] RetaGNN: Relational temporal attentive graph neural networks for holistic sequential recommendation, WWW2021.
>
> [R5] Self-supervised hypergraph convolutional networks for session-based recommendation, AAAI2021.

---

### Official Review · Reviewer_YVVZ · 2022-07-10

**Rating:** 7
**Confidence:** 3
**Soundness:** 3 good
**Presentation:** 3 good
**Contribution:** 3 good

**Summary:**

This paper aims to present a theoretical analysis of GCN-based recommender systems, and proposes a method that balances empirical risk and model complexity. The authors first utilize the Covering number to obtain the Covering number bound for GCN-based models over bipartite graphs. Then, they derive two generalization guarantees for both inductive learning and transductive learning. Furthermore, they propose an algorithm, Item Mixture (IMix), that can improve the performance of recommender systems by mixing up two positive-negative item pairs. At last, they conduct expensive experiments to demonstrate the effectiveness of the proposed ideas.

**Questions:**

1. Can the proposed method be extended to other GCN-based tasks?

2. IMix mixes item embeddings but not user embeddings. Are there any specific reasons?

3. In the experiments, Xavier is used to initialize node features. Therefore, the model highly relies on graph structure. I’m wondering what the results would be if you include existing node features.


**Limitations:**

GCN-based recommender system is a relatively narrow direction for recommender systems. As the authors mentioned in their future work, it would be more applicable if they extend their work to GNN-based or knowledge graph-based systems.

**Strengths And Weaknesses:**

Strengths

+The research problem studied in this paper is very interesting. The idea of using the Covering number to derive generalization bounds to GCNs could provide a new direction for further development.

+The derivation of theoretical analysis (generalization bounds and proof) is clear.

+The empirical results look good and consistent with the analysis.

Weaknesses

-The connection between the generalization bounds and Item Mixture (IMix) is not strong enough. The goal of IMix is to balance empirical error and model complexity. However, the motivation of IMix is more like a regularization loss to alleviate the overfitting than an algorithm to find an optimal solution for both reducing model complexity and improving recommendations.

-The typical methods to measure model complexity are Vapnik-Chervonenkis Dimension (VC-Dimension) and Rademacher complexity. Why do the authors use the Covering number? It would be better if they provide a discussion about the advantages of the Covering number compared to those two measurements.

-The theoretical findings include transductive learning and inductive learning. However, in the experiments, the authors do not design specific tests to compare these two settings.

---

> ### Author Response · Authors · 2022-08-02
> **Response to Reviewer YVVZ: Part 3**
>
> ### Q7. It would be applicable to extend the work to GNN-based or knwoledge graph-based systems.
>
> Thanks for your valuable suggestion. Referring to our response to Q4, both theoretical analysis and IMix have the potential to be extended to other systems. We look forward to generalizing them to develop more connections between graph-based recommender systems.
>
> [R1] Advances in collaborative filtering, Recommender systems handbook.

---

> ### Author Response · Authors · 2022-08-02
> **Response to Reviewer YVVZ: Part 2**
>
> ### Q4. Can the proposed method be extended?
>
> The answer is “Yes”. We will illustrate the scalability of the proposed method from two aspects.
>
> **For the theoretical analysis**, we first provide Covering number-based generalization bounds for inductive and transductive learning (**Lemma 2** and **Lemma 3**), which can be generalized to other tasks with bounded loss functions. In the model complexity analysis, we recursively link the output spaces of the recommendation models and each layer of GCNs via $l_2$-Covering of the parameters. Similar proof strategy can be extended to other downstream tasks that rely on node representation learning.
>
> **For the proposed IMix,** we can strictly guarantee the effectiveness from empirical and theoretical aspects for GCN-based recommendations. Theoretically, it could work as long as the model is inference based on embeddings and trained with bounded ranking loss. We also empirically verify its potential in the non-GCN-based model in Appendix A.3. However, due to models' diversity, broad applicability is hard to guarantee, which will be our future work.
>
> ### Q5. Why does IMix only mix item embeddings?
>
> We have added additional experiments about mixing both user embedding and item embedding (denoted as UIMix) in the revised version (Appendix A.8), and some results are described below.  Note that we do not consider the version that only mixes user embeddings, because it must ensure that the positive and negative items are the same, i.e., $e_i=e_j$ and $e_{i'}=e_{j'}$, otherwise, a quintuple $(\tilde{e_u},e_i,e_{i'},e_j,e_{j'})$ instead of the recommendation triplet $(\tilde{e_u},e_i,e_{i'})$, which does not conform the model training. we find that the performance of UIMix is not as good as IMix. One possible reason is that the recommender system has two types of feature spaces: user space and item space. IMix mixes in the item space of the same user (same user space), while UIMix mixes across feature spaces (different user space and different item space), which may lead to confusion in learning. Please refer to Appendix A.8 for more discussion and results.
>
> | Methods | Recall@20(Gowalla) | NDCG@20(Gowalla) | Recall@20(Yelp2018) | NDCG@20(Yelp2018) |
> | --- | --- | --- | --- | --- |
> | LightGCN | 0.1556 | 0.1340 | 0.0526 | 0.0427 |
> | LightGCN-UIMix | 0.1646 | 0.1394 | 0.0558 | 0.0457 |
> | **LightGCN-IMix** | **0.1699** | **0.1436** | **0.0563** | **0.0458** |
> | NGCF | 0.1308 | 0.1031 | 0.0419 | 0.0336 |
> | NGCF-UIMix | 0.1294 | 0.1079 | 0.0437 | 0.0352 |
> | **NGCF-IMix** | **0.1451** | **0.1146** | **0.0492** | **0.0360** |
>
> ### Q6. What would be the result using the existing node features instead of initialization?
>
> In collaborative filtering, due to the lack of rich features, random embeddings are generally used as the initial features and learned during training, which is especially common in GCN-based recommendation [2, 7-9, 39]. Nonetheless, we explore whether recommendation models using raw features can achieve similar findings compared with standard models. To solve the problem of misalignment of user and item feature dimensions, we use a Multilayer Perceptron to map them into the same dimensional space. We denote NGCF and LightGCN using node features as F-NGCF and F-LightGCN. The numerical discussion can be found in Fig.15 of Appendix A.9 (the revised version). We find that the effect of graph normalization in the node-features-included model (Fig.15a) is much smaller than the standard model that does not use the node features (Fig.2a). This may be that the node features weaken the high dependence on the graph structure, which is consistent with the reviewer’s insight. But it is undeniable that our findings still hold that using graph normalization helps model generalization.
>
> Besides, we apply IMix to node-features-included models (F-NGCF and F-LightGCN), and denote them as F-NGCF-IMix and F-LightGCN-IMix. The results are shown below. Obviously, the performance equipped IMix has been gratifyingly improved, and the maximum improvement is an astonishing 24.85% concerning Recall@20.
>
> Furthermore, compared with the table in the response of Q5, we find that the performance of node-feature-included models is worse. We reasonably suspect that the datasets lack rich features (Gowalla has no node features, while Yelp2018 only has about 1/3 of data with 17-dimensional features. One-hot encoding is adopted for missing features), which limits the model's expressiveness. More discussion and results can be found in Appendix A.9.
>
> | Method | Recall@20(Gowalla) | NDCG@20(Gowalla) | Recall@20(Yelp2018) | NDCG@20(Yelp2018) |
> | --- | --- | --- | --- | --- |
> | F-NGCF | 0.1060 | 0.0820 | 0.0338 | 0.0268 |
> | **F-NGCF-IMix** | **0.1217(+14.81%)** | **0.0965(+17.69%)** | **0.0422(+24.85%)** | **0.0337(+25.75%)** |
> | F-LightGCN | 0.1334 | 0.1124 | 0.0445 | 0.0361 |
> | **F-LightGCN-IMix** | **0.1552(+16.34%)** | **0.1334(+18.68%)** | **0.0530(+19.10%)** | **0.0435(+20.50%)** |

---

> ### Author Response · Authors · 2022-08-02
> **Response to Reviewer YVVZ: Part 1**
>
> Many thanks to Reviewer YVVZ for positive comments and constructive feedback.
>
> ### Q1. The connection between the generalization bounds and IMix is not strong enough.
>
> Thank you for this insightful question. As shown in Theorem 1, the generalization error of the GCN-based model mainly depends on the empirical risk ($\hat{\mathcal{L}}_m(f)$ in RHS) and model complexity (middle parts in RHS). In existing studies, most models bring excessive complexity in pursuit of high expressiveness (low empirical risk), while simple models cannot guarantee low empirical risk. To this end, the design of IMix is based on preserving expressive power (without modifying the model structure or hyperparameters) while reducing the model complexity (proved in **Remark 3**).
>
> For the reviewer’s confusion about the relationship between IMix and regularization loss, we proved that IMix training loss is approximately equivalent to standard loss with a regularization term (**Lemma 4**). Therefore, from the perspective of model loss, the reviewer’s opinion is correct. When further considering the generalization error, the regularization term included by IMix is essentially reducing the model complexity (**Remark 3**). Therefore, **from the perspective of model loss**, IMix adds a regularization term, while **from the perspective of the generalization error**, it is the reduction of model complexity emphasized in the paper. Their essential purpose is to improve generalization performance. Notably, IMix’s adaptive regularization term is inherently contained and does not impose any constraints on models, which is different from ordinary regularization loss.
>
> ### Q2. Why do authors use Covering number?
>
> Thanks for your insightful questions. Before answering, we must expose two major challenges that hinder theoretical analysis for GCN-based recommender systems as follows.
>
> - There are high dependencies between data.
> - Non-linear activations lead to non-convex output space.
>
> Rademacher complexity is a measurement that highly depends on data distribution and hypothesis space. Even given the data distribution, it is difficult for Rademacher complexity to estimate the upper bound of the output due to the non-convex output space. In addition, for VC-dimension, although it does not consider the output bound, it estimates model complexity by shattering all samples (i.e., classifying all samples correctly), which may result in overly pessimistic and trivial results [22].
>
> However, Covering number gives us much freedom for customization: choice of metric and discretization $^1$. Specifically, the Covering number used in this paper is with respect to the infinity norm ($l_{\infty}$-norm), and the $l_{\infty}$-Covering allows us to study data-independent model complexity. In addition, the essence of Covering number is the discretization of rich function classes. This characteristic allows us to recursively investigate the complexity of deep non-linear GCNs with discretization resolution, then a tight bound can be yielded via chaining.
>
> In addition, as the analysis using $l_{\infty}$-Covering number is in the worst-case sense, one of the future directions is to explore data-dependent complexity using  $l_1$ or  $l_2$-norm.
>
> $^1$  [notes.pdf (stanford.edu)](https://web.stanford.edu/class/cs229t/2015/notes.pdf)
>
> ### Q3. This paper does not provide specific tests for transductive and inductive learning.
>
> We have added Pin-Sage based on inductive learning and  compared it with NGCF and LightGCN in **Appendix A.5 (revised version)**. Some details are described in the table below (1 layer model concerning Recall@20).  We find that the performance of Pin-Sage is inferior to LightGCN and NGCF. We reasonably suspect that in collaborative filtering, the recommendation results mainly depend on historical interactions rather than  node features [R1], which reflects that the performance of GCN-based models is more dependent on its neighbors. However, Pin-Sage uses neighbor sampling to ensure the model's flexibility and generalization, which loses lots of valuable information and damages the recommendation performance. See **Appendix A.5** for more results and discussion.
> | Methods  | Gowalla | Yelp2018 |
> | --- | --- | --- |
> | Pin-Sage | 0.1003 | 0.0396 |
> | NGCF | 0.1308 | 0.0419 |
> | LightGCN | 0.1556 | 0.0526 |

---

### Official Review · Reviewer_UAWr · 2022-07-28

**Rating:** 6
**Confidence:** 3
**Soundness:** 3 good
**Presentation:** 2 fair
**Contribution:** 2 fair

**Summary:**

Note: I was called upon as an emergency review and only get to check the paper briefly.

This paper proposes to study the generalization ability of GNNs for recommender systems (i.e., bi-partite graphs). Specifically, the authors establish the generalization guarantee by deriving the covering number. Based on the results, the authors propose the Item Mixture (IMix) method to enhance recommendation using the idea of Mixup in self-supervised learning. Experimental results demonstrate the effectiveness of the proposed method.


**Questions:**

See above

**Limitations:**

N.A.

**Strengths And Weaknesses:**

Pros:
(+) Studying the generalization ability of GNNs from the covering number perspective seems unexplored and interesting.
(+) The proposed IMix method is intuitive to understand yet enjoy theoretical guarantees.

Cons and concerns:
(-) My primary concern is how the proposed analyses are related to recommender systems. Specifically, the GNN formulation in Eq. (2) seems to fit any graph beside the recommendation scenario (i.e., bi-partite graphs). If that is the case, should the results in Section 3 generalize to other GNNs? On the other hand, can the existing analyses for general GNNs (ref [17-24]) apply to bi-partite graphs? If yes, how are their results compared to this paper?

(-) Though very recent, mixup-based techniques have been introduced in GNNs, which should be discussed.
Mixup for Node and Graph Classification, WWW 2021
G-Mixup Graph Data Augmentation for Graph Classification, ICML 2022

(-) Experiments could be enhanced considering only two datasets (Gowalla and Yelp2018) and two baselines (NGCF and LightGCN) are adopted.

(-) The organization and presentation of the paper could be improved. For example, there are confusing/unexplained sentences/symbols and inaccurate expressions (as detailed below). Besides, I would suggest separating related works into another section from the introduction.
(1)  What is \hat{R}_u and \hat{R}_i (L81, P2)?
(2)  The definition of transductive and inductive learning is confusing. In L133, the authors write, "The testing samples in inductive learning are drawn from some unknown distribution," while in L138, "we assume that all samples in Sm are i.i.d. according to D, denoted as Sm ∼ Dm". Are D and D^M the same? If not, how could the generalization error be bounded for an unknown distribution? (violating the no free lunch theorem?)
(3)  Some superscripts have parenthesis while others do not.
(4)  L188: "In addition, one recent work [33] gives a new theoretical finding that non-linearity does not contribute to improving expressive power of GNNs," which may not be accurate since [33] focuses on the over-smoothing issue of deep GNNs. Based on "How Neural Networks Extrapolate: From Feedforward to Graph Neural Networks, ICLR 2021", non-linear activation functions are critical for GNNs to generalize.

---

> ### Author Response · Authors · 2022-08-02
> **Response to Reviewer UAWr: Part 3**
>
> ### Q4. The organization and presentation of the paper could be improved.
>
> > **(1) What is \hat{R}_u and \hat{R}_i (L81, P2)?**
> >
>
> Similar to related GCN-based recommendation works [8, 9], we denote the user-item interaction matrix as $\mathbf{R}$ and the adjacency matrix as $\mathbf{A}=\left [ \begin{matrix}0 & \mathbf{R}  \\\ \mathbf{R^T} & 0 \end{matrix} \right ]$. Then we denote the corresponding normalized graph filter of $\mathbf{A}$ as $\tilde{\mathbf{A}}$, **which is still a block anti-diagonal matrix** for different graph normalizations. Therefore, we denote the two non-zero blocks in $\tilde{\mathbf{A}}$$\mathbf{}$ as $\tilde{\mathbf{R}}_u$ and $\tilde{\mathbf{R}}^T_i$, i.e., $\tilde{\mathbf{A}}=\left [ \begin{matrix}0 & \tilde{\mathbf{R}}_u  \\\ \tilde{\mathbf{R}}^T_i & 0 \end{matrix} \right ]$.
>
> We have added the descriptions of these matrices in the revised paper (**Section 2.2**).
>
> > **(2) The definition of transductive and inductive learning is confusing. In L138, are D and D^M the same?**
> >
>
> In the paper, $\mathcal{D}$ and $\mathcal{D}^m$ denote the **same fixed but unknown distribution**. The definition of $\mathcal{D}^m$ follows the standard format in machine learning theory [26], which marks the number of sampled data $m$ on distribution $\mathcal{D}$, i.e., **$\mathcal{D}^m$ is just another representation of $\mathcal{D}$ when sampling**,  and $S_m\sim \mathcal{D}^m$ means $m$ samples in training set $S_m$ are drawn i.i.d. according to $\mathcal{D}$. Therefore, “The testing samples in inductive learning are drawn from some unknown distribution” and “we assume that all samples in Sm are i.i.d. according to D, denoted as Sm ∼ Dm” are **equivalent,** and both mean that the samples are sampled from the same distribution $\mathcal{D}$.
>
> > **(3) Some superscripts have parenthesis while others do not.**
> >
>
> Thanks for your comment, we have unified all superscript formats in the revised paper (e.g., Eq. 1 and Eq. 2).
>
> > (**4)The expression in line 188 about [33] may not be accurate. And based on [R4], non-linear activation functions are critical for GNNs to generalize.**
> >
>
> Thanks for your comment. We have rephrased the citation for a more accurate and clear expression in the revised paper. We cite [33] to state that non-linear functions **do not bring considerable expressive power** to learn better node features, but negatively **increase the difficulty of model training** [9]. This is consistent with the statement on Paper 2 of  [33] that under certain conditions, neither layer stacking nor non-linearity contributes to improving expressive power, different from other DNNs (e.g., FNNs and CNNs).
>
> For the recent work [R4] commented by the reviewer, they investigated the **extrapolation performance** of GNNs, while our work focuses on **in-distribution generalization** (**i.e., interpolation performance**). Quite different tasks make their results may not be suitable for our work.
>
> Besides, we emphasize that Theorem 1 reveals non-linear activations present negligible influence on generalization performance. In addition to the support of our extensive experiments (Fig 2b, Fig 6b, Fig. 11b, Fig. 12b, Fig. 13b in the revised paper), recent works have also met similar findings [9, R3].
>
> [R1] Mixup for node and graph classification, WWW 2021.
>
> [R2] G-Mixup: Graph Data Augmentation for Graph Classification, ICML 2022.
>
> [R3] Simplifying graph convolutional networks, ICML 2019.
>
> [R4] How neural networks extrapolate: From feedforward to graph neural networks, ICLR 2021.

---

> > ### Comment · Reviewer_UAWr · 2022-08-04
> > **Response for Rebuttal**
> >
> > I appreciate the authors' efforts in the rebuttal, including more experimental results, clarifications, and paper revisions. The responses have addressed most of my concerns, and I have improved my score accordingly. A minor comment is that the authors may want to further clarify that this paper focuses on in-distribution generalization instead of out-of-distribution generalization, which led to some of my confusion, such as Q4(2)(4) in the first place.

---

> > > ### Author Response · Authors · 2022-08-06
> > > **Response to Reviewer UAWr's Additional Comment**
> > >
> > > Thanks for your approval of our work and enlightening comments. It seems that the distributions regarding inductive and transductive studied in this paper have confused you. We will give clearer definitions below and rephrase the paper accordingly (**Section 3.2**).
> > >
> > > **Inductive learning [26].** We assume that all samples are i.i.d. according to a fixed but unknown distribution $\\mathcal{D}$. The model receives a training set $S\_m=\\{x\_i, y\_i \\}\_{i=1}^m$ drawn i.i.d. according to $\\mathcal{D}$, and aims at making accurate prediction about any unobserved samples $(x, y) \\sim \\mathcal{D}$. Then the empirical error and generalization error can be formalized as follows.
> > >
> > > $$
> > > \\begin{aligned}& \\text{Empirical error: } & \\hat{\\mathcal{L}}\_m(f) = \\frac{1}{m} \\sum\\nolimits\_{(x\_i, y\_i)^\\in S\_m} l(x\_i, y\_i) \\\ & \\text{Generalization error: } & {\\mathcal{L}}(f) = \\mathbb{E}\_{(x, y)\\sim \\mathcal D} l(x, y)\\end{aligned}
> > > $$
> > >
> > > **Transductive learning [30].** Given a fixed set $S\_{m+u}=\\{x\_i, y\_i \\}\_{i=1}^{m+u}$, we randomly partition $S\_{m+u}$ into two disjoint sets of $m$ and $u$ samples, i.e., training set $S\_m$ and testing set $S\_u$. Let $X\_u = \\{x\_i \\}\_{i=1}^u$ corresponding to $S\_u$, the model aims at predicting the labels of testing set based on $S\_m \\cup X\_u$. Then the empirical error and generalization error can be formalized as follows.
> > >
> > > $$
> > > \\begin{aligned}& \\text{Empirical error: } & \\hat{\\mathcal{L}}\_m(f) = \\frac{1}{m} \\sum\\nolimits\_{(x\_i, y\_i)^\\in S\_m} l(x\_i, y\_i) \\\ & \\text{Generalization error: } & {\\mathcal{L}}\_u(f) = \\frac{1}{u} \\sum\\nolimits\_{(x\_i, y\_i)^\\in S\_u} l(x\_i, y\_i) \\end{aligned}
> > > $$
> > >
> > > As defined above, for inductive learning we focus on in-distribution generalization, which means all samples are i.i.d. drawn from a fixed but unknown distribution. Transductive learning has observed all the data beforehand and learns from the already observed training set to predict the labels of the testing set. The training and testing sets are obtained via a random split (i.e., uniformly sampling without replacement) from a fixed and finite set, so transductive learning does not focus on out-of-distribution generalization and is essentially distribution-free [30]. Notably, the dependency of the training set and test set enables us to analyze generalization performance. Thanks again for your impressively insightful review, which greatly helps us in paper revision.

---

> ### Author Response · Authors · 2022-08-02
> **Response to Reviewer UAWr: Part 2**
>
> ### Q2. Some mixup-based GNNs should be discussed.
>
> As per your request, we have supplemented discussions of related mixup-based GNNs [R1, R2] as below:
>
> - The work [R1] used two-branch graph convolution operations, where one is applied to the original features and the other one to the mixed features, and they also performed Mixup at each layer. In contrast, our proposed IMix does not introduce additional convolution branches and only performs Mixup at the last layer. In addition, we provide theoretical guarantees for the proposed strategy, **which are lacking in [R1].**
> - G-Mixup [R2] is a graph-level augmentation method, which is different from our recommendation task (i.e., node level). Besides, G-Mixup enhances the model by interpolating the graphon of different classes of graph, but we only mix the item embeddings of the same users. In the revised paper, we have supplemented more discussion and experiments of these related works.
>
> Notably,  we have **conducted comparitive experiments with two-branch Mixup (TBMix) [R1],** and the experimental results of LightGCN configured IMix and TBMix with respect to Recall@20 are as follows.
>
> | layer | Methods | Gowalla | Yelp2018 |
> | --- | --- | --- | --- |
> | 1 layer | LightGCN | 0.1556 | 0.0526 |
> | 1 layer | LightGCN-TBMix | 0.1495 | **0.0585** |
> | 1 layer | **LightGCN-IMix** | **0.1699** | 0.0562 |
> | 2 layer | LightGCN | 0.1672 | 0.0564 |
> | 2 layer | LightGCN-TBMix | 0.1461 | 0.0595 |
> | 2 layer | **LightGCN-IMix** | **0.1765** | **0.0620** |
>
> As shown above, our proposed IMix outperforms TBMix on Gowalla dataset and Yelp2018 dataset in most cases, and significantly improves the performance of LightGCN. Although TBMix performs well on Yelp2018, it performs poorly on Gowalla, even inferior to LightGCN. More experiments (different layers, NDCG@20) have been added in Appendix A.8 (in the revised paper).
>
> ### Q3. Experiments could be enhanced.
>
> Thank you for this suggestion. We have added supplemented extensive experiments in Appendix A (the revised paper). Specifically, they include the newly added LastFM dataset (Section A.7), two newly added GCN-based baselines (Section A.6), the comparison between transductive learning and inductive learning (Section A.5), the comparison with the recent mixup-based strategies (Section A.8), the node-features-included models (Section A.9) and the visualization of IMix (Section A.10). These added models all include numerical discussion and effectiveness analysis of IMix.
>
> We show partial results of IMix on the LastFM dataset (NGCF model, more results have been added in **A.7**) and GCMC and SpectralCF (denoted as GCMC-IMix and SpectralCF-IMix. They are evaluated on Gowalla dataset, and more results have been added in **A.6**)
>
> | LastFM | Method | Recall@20 | NDCG@20 |
> | --- | --- | --- | --- |
> | 1 layer | without IMix | 0.2341 | 0.1746 |
> | 1 layer | **with IMix** | **0.2635(+12.50%)** | **0.1986(+13.76%)** |
> | 2 layer | without IMix | 0.2299 | 0.1714 |
> | 2 layer | **with IMix** | **0.2630(+14.41%)** | **0.1991(+16.15%)** |
>
> | Method | Recall@20(Gowalla) | NDCG@20(Gowalla) | Recall@20(Yelp2018) | NDCG@20(Yelp2018) |
> | --- | --- | --- | --- | --- |
> | GCMC | 0.1328 | 0.1142 | 0.0484 | 0.0387 |
> | **GCMC-IMix** | **0.1491(+12.27%)** | **0.1201(+5.16%)** | **0.0546(+12.81%)** | **0.0441(+13.95%)** |
> | SpectralCF | 0.1424 | 0.1171 | 0.0496 | 0.0400 |
> | **SpectralCF-IMix** | **0.1588(+11.52%)** | **0.1314(+12.21%)** | **0.0546(+10.08%)** | **0.0443(+10.75%)** |

---

> ### Author Response · Authors · 2022-08-02
> **Response to Reviewer UAWr: Part 1**
>
> We sincerely thank Reviewer UAWr for thorough and insightful comments.
>
> ### Q1. How are the proposed analyses related to recommender system?
>
> Our theoretical analysis includes complexity-based generalization error and Covering number-based complexity. We will state the relationships between the recommendation task and the analysis in these two parts:
>
> - **Generalization analysis.** We focus on the ranking loss in recommender systems and link the discrete Covering number of function class and the generalization errors for inductive and transductive learning (**Lemma 2** and **Lemma 3**). To extend the bounds to GCN-based recommendation, we establish the connections between the recommendation model output space and the GCNs output space (**Proof of Lemma 1: Step 3**). Here the output space of GCNs means the learned representation space.
>
>     To the best of our knowledge, the existing theoretical studies for GNNs exploring similar tasks (i.e., link prediction) only consider the final node embeddings and do not further investigate specific models [23, 24].
>
> - **Complexity analysis.** The key technique of this paper is $l_{\infty}$-Covering number with discretization resolutions (i.e., radius of balls).  Depending on $l_{\infty}$-norm and the matrix Covering of parameters, we recursively analyze the complexity of the output space at each layer. It should be noted that we consider the user representation space and item representation space separately when handling recursion (**Proof of Lemma 1: Step 1&Step 2**).
>
> > **(1) Should the results generalize to other GNNs?**
> >
>
> The answer for final generalization results is “No”, but “Yes” for theoretical analysis. Since **ordinary graphs lack the bipartite graph properties** (i.e., pairwise computation between different parts and information propagation between two parts), the same task is **simpler on more general ordinary graphs**. We can only relax our analysis to consider the output space of all nodes simultaneously, but this **brings a looser bound to the recommendation scenario** (i.e., bipartite graphs).
>
> Therefore, the tighter generalization bound we obtained cannot generalize to other GNNs, but the theoretical analysis after relaxation can. We very much look forward to generalizing the theoretical analysis to other GNNs to provide critical insights for formulating more advanced models, which will serve as our future work.
>
> > **(2) Can the existing analyses for general GNNs apply to bi-partite graphs?**
> >
>
> First, recommendation and node-level tasks are different, and even if the first problem is solved, extending existing analyses to bipartite graphs can only lead to looser generalization bounds than ours. specifically,
>
> - **Different types of tasks.** The existing studies focused on node (or graph) classification tasks, only involving a single node (or the sum of all nodes). However, our work studies the ranking task in recommender systems, and the output space relates to pairwise nodes. Function class, including pairwise computation, may be more complicated and bring challenges for handling recursion in multi-layer GNNs, which is one of our technical contributions.
> - **Bipartite graphs vs. original graphs.** Similar to the response to (1), the existing analyses need to relax the bipartite graph properties, which make it impossible for them to consider the representation spaces of users and items separately like our work. Therefore, they will get a looser bound for some tasks on bipartite graphs.

---

### Meta-Review · Area_Chair_bDKR · 2022-08-25

**Recommendation:** Accept
**Confidence:** Certain

**Metareview:**

This paper proposes a method named Item Mixture (IMix) for recommendation systems. The proposed method is based on Mixup techniques and can enhance the generalization ability with theoretical guarantees. Some concerns regarding the relevance with respect to recommendation systems and the clarity of the paper have been initially raised, but have been addressed during the rebuttal. Now that the reviewers are uniformly positive, I recommend acceptance and encourage the authors to make necessary modifications to incorporate the additional contents of the rebuttal into the main paper.

**Award:**

No

---

### Decision · Program_Chairs · 2022-09-14

Accept